# Uveal Melanoma, Angiogenesis and Immunotherapy, Is There Any Hope?

**DOI:** 10.3390/cancers11060834

**Published:** 2019-06-17

**Authors:** Florian Castet, Sandra Garcia-Mulero, Rebeca Sanz-Pamplona, Andres Cuellar, Oriol Casanovas, Josep Maria Caminal, Josep Maria Piulats

**Affiliations:** 1Medical Oncology Department, Catalan Institute of Cancer (ICO), IDIBELL-OncoBell, Hospitalet de Llobregat, 08908 Barcelona, Spain; fcastet@iconcologia.net (F.C.); macuellar@iconcologia.net (A.C.); 2Clinical Research in Solid Tumors Group (CREST), Bellvitge Biomedical Research Institute IDIBELL-OncoBell, Hospitalet de Llobregat, 08908 Barcelona, Spain; s.garciam@idibell.cat; 3Unit of Biomarkers and Susceptibility, Cancer Prevention and Control Program, Catalan Institute of Oncology (ICO), IDIBELL-OncoBell, Hospitalet de Llobregat, 08908 Barcelona, Spain; rebecasanz@iconcologia.net; 4Tumor Angiogenesis Group, ProCURE, Catalan Institute of Oncology, IDIBELL-OncoBell, L’Hospitalet de Llobregat, 08908 Barcelona, Spain; ocasanovas@iconcologia.net; 5Ophthalmology Department; University Hospital of Bellvitge, IDIBELL, Hospitalet de Llobregat, 08907 Barcelona, Spain; jmcaminal@gmail.com; 6Centro de Investigación Biomédica en Red de Cáncer (CIBERONC), 28029 Madrid, Spain

**Keywords:** uveal melanoma, angiogenesis, immunotherapy, tumour microenvironment, vasculogenic mimicry

## Abstract

Uveal melanoma is considered a rare disease but it is the most common intraocular malignancy in adults. Local treatments are effective, but the systemic recurrence rate is unacceptably high. Moreover, once metastasis have developed the prognosis is poor, with a 5-year survival rate of less than 5%, and systemic therapies, including immunotherapy, have rendered poor results. The tumour biology is complex, but angiogenesis is a highly important pathway in these tumours. Vasculogenic mimicry, the ability of melanomas to generate vascular channels independently of endothelial cells, could play an important role, but no effective therapy targeting this process has been developed so far. Angiogenesis modulates the tumour microenvironment of melanomas, and a close interplay is established between them. Therefore, combining immune strategies with drugs targeting angiogenesis offers a new therapeutic paradigm. In preclinical studies, these approaches effectively target these tumours, and a phase I clinical study has shown encouraging results in cutaneous melanomas. In this review, we will discuss the importance of angiogenesis in uveal melanoma, with a special focus on vasculogenic mimicry, and describe the interplay between angiogenesis and the tumour microenvironment. In addition, we will suggest future therapeutic approaches based on these observations and mention ways in which to potentially enhance current treatments.

## 1. Introduction

Uveal melanoma (UM) is a malignant tumour that arises in the melanocytes located in the uveal tract [1]. In 85–90% of cases the choroid is involved, while in the remaining 10–15% the tumour will arise in the iris and ciliary body [2,3,4]. It is considered a rare cancer, with an estimated incidence of 4.9–5.2 cases per million in the United States [5], which has remained stable in the past 20 years [6]. In Europe, the incidence seems to be highest in the Northern countries, with an incidence rate of more than 8 cases per million in Norway and Denmark [7]. However, in Southern countries, such as Spain or Italy, the incidence is less than 2 cases per million. This pattern is interestingly in contrast to what is observed in cutaneous malignant melanoma, probably due to phenotypic and racial disparities [8].

Over 95% of patients have disease confined to the eye upon diagnosis [9], mostly owing to its early clinical manifestations [10]. However, up to 50% of patients will develop metastasis [4], mostly to the liver [9]. Many factors have been associated to an increased risk of metastasis, including clinical factors such as tumour thickness, ciliary body location, or increasing tumour diameter [4]. More recently, molecular and cytological factors such as monosomy of chromosome 3, additional copies of chromosome 8 [11], and *BRCA1* associated protein (*BAP1*) mutations [12], have been related to decreased survival. On the other hand, Splicing factor 3B subunit 1 (*SF3B1*) and Eukaryotic translation initiation factor 1A (*EIF1AX*) mutations seem to confer improved survival [13,14,15,16].

The prognosis of metastatic UM remains dismal, with a median overall survival (OS) of less than a year in most cases [1,17,18]. Moreover, survival rates seem to have remained stable over the past 40 years, reflecting the lack of current effective systemic strategies [1,19,20]. Indeed, the objective response rates (ORRs) of commonly used chemotherapies are unacceptably low [21]. Single-agent chemotherapies, such as fotemustine [22] or dacarbacine [23], have ORRs of 2.4% and 8%, respectively, with median progression-free survivals of less than 3 months in both cases. In addition, combined chemotherapy regimens, such as dacarbazine–treosulfan [24] or gemcitabine–treosulfan [25] have also shown disappointing results, with ORRs of 0% and 4.2%, respectively.

The emergence of immunotherapy has changed the natural history of cutaneous malignant melanoma. Indeed, anti-programmed death ligand 1 (PDL1) and anti-programmed death 1 (PD1) antibody monotherapy have shown to significantly improve survival in metastatic cutaneous melanoma compared to standard chemotherapy [5,26,27]. However, the most intriguing question of immunotherapy in this tumour is whether the durable complete responses observed with these treatments could translate into a possibility for cure, in a clinical setting which was previously considered incurable [28]. In a pooled-analysis of more than 1800 patients with advanced cutaneous melanoma treated with ipilimumab, an anti-cytotoxic T-lymphocyte antigen-4 (CTLA-4) monoclonal antibody, the survival curve seemed to plateau at 21% starting at year 3, implying that one-fifth of patients could achieve long-term survival and could eventually be cured with immunotherapy alone [29]. Longer follow-up of the above-mentioned anti-PD1 and anti-PDL1 trials are eagerly awaited to study this phenomenon further. Moreover, combining anti-CTLA4 and anti-PD1 therapies has proved to significantly increase survival compared to anti-PD1 monotherapy [30]. More interestingly so, the survival curves seem to plateau at more than 50% [31], although this will require further confirmation with longer follow-up.

However, these encouraging results have not been reproduced in UM. In two phase II trials that studied the role of ipilimumab in metastatic UM, the ORR varied from 0% to 7.7%, with an OS of 6.8 months in this population [32,33]. Anti-PD1 and anti-PDL1 did not increase efficacy in this patient population, with an ORR of 3.6% and a median OS of 7.6 months [34]. Lastly, combining both antibodies rendered a disappointing ORR of 12%, with a median OS of 12.7 months [35].

UM develops in one of the most capillary-rich tissues of the body and is disseminated haematogenously. Most UM cell lines, but not normal melanocytes, strongly synthesize and secrete vascular endothelial growth factor (VEGF) and basic fibroblast growth factor (bFGF) during cell culture [36]. As we will thoroughly discuss, vascular abnormalities facilitate immune evasion [37] and angiogenic signatures are frequently enriched in tumours with resistance to checkpoint inhibitors [38]. Indeed, VEGF and other angiogenic factors play an important role in modulating the immune system directly by suppressing dendritic cell maturation [39], inhibiting T-cell effector response [40], and recruiting myeloid derived suppressor cells [41].

These assumptions suggest that a combined therapeutic approach of immunotherapy and antiangiogenic drugs could potentially overcome the adverse microenvironment of UM, and therefore increase the efficacy of checkpoint inhibitors.

## 2. Angiogenesis in Melanoma

Angiogenesis is a hallmark of cancer [42], and is defined as the development of new blood vessels from a pre-existing vascular network in order to supply the nutritional and metabolic demands of tumours [43]. Progression of benign lesions to malignant tumours often requires an angiogenic switch [44], understood as a time-restricted event during tumour progression where the balance between pro- (such as vascular endothelial growth factor-A (VEGF-A)) and anti-angiogenic factors (such as thrombospondin-1 (TSP-1)) tilts towards a pro-angiogenic outcome [45], eventually leading to the formation of new blood vessels [43]. Due to the chronic activation of pro-angiogenic factors, the resulting vessels often acquire an aberrant morphology [42,43].

In addition to endothelial cells, pericytes seem to play an essential role in regulating vessel maturation, stabilization, quiescence and function in cancer [46]. Pericytes are branched, contractile cells present in capillaries that physiologically regulate microvascular blood flow [47]. They are believed to contribute to the aberrant morphology of the resulting vessels [48]. Furthermore, pericytes seem to regulate cell metastasis. Indeed, pericyte dysfunction results in an increase in distant metastasis [49,50]. Factors involved in regulating their functions include platelet-derived growth factor-B (PDGF-B) [51], angiopoietin 1 and 2 (Ang1 and Ang2) [52], transforming growth factor B (TGF-B) [47], and vascular endothelial growth factor (VEGF) [52].

Finally, apart from endothelial cells and pericytes, macrophages seem to be major contributors to angiogenesis. In 2001, tumour infiltrating macrophages (TAMs) were found to be associated to worse prognosis in UM [53], and Bronkhorst et al. demonstrated in 2010 that this was mainly due to polarized M2 macrophages [54]. These macrophages are associated to immunosuppressive functions and are proangiogenic cells [55,56,57]. Interestingly, Bronkhorst et al. found that a high infiltration of M2 macrophages was correlated to monosomy 3 and increased microvascular density in UM [54]. Additionally, macrophages present in choroidal neovascularization specimens express VEGF, which further supports a role of these cells in UM angiogenesis [58].

Angiogenesis has been shown to be an essential feature in the transition from radial to vertical growth in cutaneous melanoma, a necessary step before invasion and metastasis [59,60,61,62]. Moreover, more vascularized tumours entail a worse prognosis [63,64], resulting in a higher risk of systemic spread and decreased survival [65]. Highly vascularized UM tumours are also more aggressive and convey a worse prognosis [66,67]. Moreover, our group has applied a well-established molecular signature for angiogenesis to the expression data available from the UM The Cancer Genome Atlas (TCGA). The “BIOCARTA_VEGF_PATHWAY” signature is composed of 29 different genes related to angiogenesis, including Hypoxia-inducible factor 1-alpha (*HIF1A*), the eukaryotic translation initiation factor (EIF), VEGFA, and von Hippel Lindau (VHL), amongst others. Tumours in the TCGA that relapse systemically show a much higher angiogenesis enrichment score than non-relapsed patients (Figure 1). Differences in disease-free survival (DFS) when comparing high vs. low angiogenesis enrichment scores (Figure 2A) were statistically significant in UM patients. However, DFS was not significant when we compared signature high vs. low using primary tumours included in the cutaneous melanoma dataset from the TCGA (Figure 2B).

The hypoxic microenvironment of these tumours leads to the release of hypoxia-inducible factor 1 (HIF-1), which in turn induces VEGF, Ang-2, matrix metalloprotease 14 (MMP14), and angiogenin [68,69,70,71]. Other important pro-angiogenic factors involved in advanced melanoma include interleukin-8 (IL-8) [72,73,74] and PDGF [61,69,75,76]. High levels of Ang2 [77], VEGF [78], IL-8, and basic fibroblast growth factor (bFGF) [79] have been correlated to poor overall survival and increased risk of recurrence [80,81], while declining levels of these factors following systemic treatment seem to correlate to response [82]. Moreover, inhibition of the HIF pathway with arylsulfonamide 64B in animal UM models results in tumour regression and improved survival [83], and inhibition of VEGF-A with bevacizumab in both mouse and human uveal melanoma inhibits the establishment of micrometastasis [84], further reflecting its importance [83].

The molecular pathways involved in angiogenesis in melanoma are complex and beyond the scope of this review [85]. Notch1 seems to be increasingly important, especially in cutaneous melanoma. Indeed, in a review of 114 primary cutaneous melanoma carried out by Murtas et al. [86], the overexpression of Notch1 in both tumour and endothelial cells was associated to microvascular density. Notch1 seems to upregulate mitogen-activated protein kinase (MAPK) through CD133, which in turn transdifferentiates into endothelial-like phenotypes, therefore promoting growth and angiogenesis [87]. In UM, however, there is no established direct relationship between the Notch signalling pathway and angiogenesis. Nevertheless, hypoxia does seem to promote growth and invasion of uveal melanoma cell lines through the activation of Notch and MAPK [88,89].

Another important role of Notch1 seems to be its involvement in vascular mimicry [90], which we will discuss in the following section.

### Vasculogenic Mimicry

Vasculogenic mimicry is referred to as the process by which aggressive melanoma cells generate vascular channels independently of endothelial cells [46]. This was first described by Maniotis et al. 1999 in cutaneous and UM tissue sections [91]. The authors observed interconnected loops of extracellular matrix containing some red blood cells, with no evidence of endothelial cells. These patterns were more common in highly invasive melanomas, whereas normal melanocytes or poorly invasive cells were unable to generate such channels [91]. In 2008, Frenkel et al. used laser scanning confocal angiography with indocyanine green to demonstrate blood circulation through leakage [92]. Since this initial description, vasculogenic mimicry has been found in many other tumours and seems to be correlated to poor tumour differentiation, lymph node involvement, distant metastasis, and TNM stage [93], and therefore entails a decreased survival [93,94].

The molecular processes and mechanisms involved in vasculogenic mimicry are elusive, and much remains to be understood [95,96]. The process is triggered by a reversion of the melanocytic tumour cells to a pluripotent embryonic-like genotype [91,97,98]. The acquisition of these stem-like properties seems to be mediated by the induction of the epithelial–mesenchymal transition (EMT), as initially observed in mammary epithelial cells [99]. Indeed, coexpression of epithelial and mesenchymal markers has been observed in cutaneous melanoma cells engaged in vasculogenic mimicry [100,101]. Additionally, a significant overexpression of EMT transcription factors involved in both the acquisition of stem cell-like properties and vasculogenic mimicry have been observed in different tumours, such as Nodal in murine melanoma and human cutaneous melanoma [102,103,104,105], Twist in hepatocellular carcinoma [106,107,108], Bcl-2 in human melanoma and hepatocellular carcinoma [108,109], Zinc-finger E-box binding homeobox (ZEB) in hepatocellular, pancreatic, and colorectal carcinoma, [110,111,112], or Snail in human breast and oral squamous cell carcinomas [113,114], amongst others. Despite this relationship not being directly established in UM, the expression of EMT-associated factors does promote invasion and growth [115], which we believe could be partly explained by their role in vasculogenic mimicry, as has been demonstrated in other tumours. Only a small subset of cells, globally termed melanoma cancer stem cells (MCSCs), that are preferentially localized in the perivascular niches of cutaneous melanomas and express stem cell markers, seem to be involved in vasculogenic mimicry [116].

MCSCs involved in the formation of tubules in cutaneous melanoma cell lines highly express vascular endothelial-cadherin (VE-cadherin) [117], a central actor in vasculogenic mimicry. Indeed, downregulation of this molecule in both cutaneous and UM lines completely abrogates vasculogenic mimicry [117,118]. VE-cadherin is induced during EMT transition [119] and upregulates transforming growth factor β (TGF-β) in breast cancer cells. Additionally, VE-cadherin is colocalized with EphA2 at cell–cell adhesions and regulates EphA2 at the cell membrane in both uveal and cutaneous melanoma lines, by mediating its ability to become phosphorylated through interactions with its membrane bound ligand, ephrin-A1 [120]. The exact molecular pathways involved in the regulation of this complex interplay remain unknown, although these findings, along with others [121] evidence the importance of intracellular phosphorylation in vasculogenic mimicry.

Microarray gene chip analysis has revealed increased expressions of laminin 5, membrane type 1-matrix metalloproteinases (MT1-MMP), and MMP-1, -2, -9, and -14 in aggressive metastatic melanoma cells compared to poorly aggressive ones, and the inhibition of the interaction between laminin-5 and MMP-2 and MT1-MMP with specific antibodies inhibits the formation of the tubular network, suggesting a strong implication of these components in vasculogenic mimicry [122]. The overexpression of MT1-MMP and MMP-2 is regulated by phosphoinositide 3-kinase (PI3K), and specific inhibitors of PI3K are able to abrogate vasculogenic mimicry in both uveal and cutaneous melanoma cells by decreasing the levels of MT1-MMP and MMP-2 [123].

Furthermore, the hypoxic microenvironment of melanomas seems to be an additional trigger of vasculogenic mimicry [124]. Mouse melanoma B16 cells implanted in the ischemic limbs of mice have more vasculogenic mimicry channels than controls, with a higher expression of HIF-1, MMP-2, MMP-9, and VEGF [125,126]. Indeed, vasculogenic mimicry seems to be partly mediated by vascular endothelial growth factor receptor 1 (VEGFR-1), as was shown by Frank et al. in ABCB5+ melanoma xenografts [127]. However, a second, VEGF-independent mechanism is also able to trigger vasculogenic mimicry, mediated by the platelet EC adhesion molecule (PECAM-1), whose expression is repressed by the neural crest specifier AP-2α [128].

The relative contribution of the aforementioned mechanisms of tumour blood supply to disease progression are unknown. However, in a study performed in mouse melanoma xenographs, mosaic vessels, vasculogenic mimicry, and endothelium-dependent vessels were observed in all stages of tumour development. However, vasculogenic mimicry seemed to be the predominant pattern in early stages of disease, while this was replaced by endothelium-dependent vessels in later stages of disease development [129]. In intraocular melanoma models, the three types of microcirculation were also observed, but endothelium-dependent vessels were more common in larger tumours while vascular mimicry seemed to be predominant in smaller lesions [130]. Therefore, vasculogenic mimicry seems to play a predominant role in early stages of disease development, both in cutaneous and UM. These observations seem to be contradicted by the initial findings of Maniotis et al. and Chang et al. [91,131], where larger tumours seemed to be richer in matrix-embedded channels. Whether vasculogenic mimicry is a time-dependent event in disease progression or simply identifies inherently more aggressive tumours is unclear.

Folberg et al. demonstrated that UM that presented vasculogenic mimicry patterns had an upregulation of genes related to differentiation and suppression of proliferation, and a downregulation of genes related to promotion of invasive and metastatic behaviour [132]. These findings are counterintuitive given the worse prognosis of melanoma patients presenting with his pattern. However, the authors hypothesize that these findings could explain the chemoresistance observed in these tumours and the late metastatic recurrences of some of these patients.

## 3. Antiangiogenic Drugs in Uveal Melanoma

Based on the preclinical evidence previously discussed, targeting angiogenesis in UM seems to be an attractive and potentially effective strategy [133]. However, results of the clinical trials that have investigated this matter have been disappointing so far (see Table 1), although they do seem to be more active than in cutaneous melanoma. Most antiangiogenic drugs render no response [134,135,136,137,138,139] and in the largest trial performed to date, the best observed response rate with sorafenib was 1.7% [140]. However, cabozantinib showed higher PFS in uveal melanoma patients compared to cutaneous melanoma [139]. This observation is based on indirect comparisons and must therefore be taken with caution.

Some antiangiogenic drugs do seem to be able to produce disease stabilization in more than 50% of patients [134,139,140,141,142], albeit for a short period of time in most cases. Moreover, one must bear in mind that the number of patients included in these trials is low, with less than 20 patients in most cases, which hinders the generalizability of the data (see Table 1). Some have questioned whether the stabilizations observed were in fact due to the antiangiogenic effects of these drugs or were simply reflecting the natural history of this disease. In order to try to answer this question, Scheulen et al. carried out the STEAM study, a randomized phase II trial, in which all patients were initially treated with sorafenib for 56 days in a run-in period. Patients with stable disease were then randomized to continue on sorafenib or placebo. The trial demonstrated a significant increase in progression-free survival (5.5 vs. 1.9 months, Hazard Ratio (HR) 0.527, *p *= 0.0079) in patients that continued on sorafenib. Cross-over to sorafenib was allowed following progression to placebo. That might explain that there were no differences observed in overall survival (OS).

Therefore, new approaches are needed in order to optimize outcomes with antiangiogenic therapies in UM.

One of the main concerns in the current approaches is that antiangiogenic strategies target molecules that are mainly involved in endothelium-mediated angiogenesis, leaving vasculogenic mimicry aside. However, as we have discussed, vasculogenic mimicry seems to be present in more aggressive melanomas, and although its biological implication is unclear, its inhibition could have a potential therapeutic role. Indeed, van der Schaft et al. demonstrated that using three angiogenic inhibitors (anginex, TNP-470 and endostatin) in human melanoma as compared to human endothelial cell lines inhibited angiogenesis in endothelial lines but did not inhibit vasculogenic mimicry in the melanoma lines. A further analysis revealed a differential expression of two endostatin receptors that could explain these observations [145]. Targeting vasculogenic mimicry, therefore, should be pursued and has been attempted with different approaches in preclinical melanoma models:
Genistein, an isoflavone present in soybeans, is able to inhibit vasculogenic mimicry in uveal melanoma C918 cell lines by reducing the expression of VE-cadherin [146].Pevonedistat, a selective and potent inhibitor of NEDD8-activating enzyme E1 subunit 1 (NAE1), an enzyme involved in neddylation, is able to repress the cancer stemness properties of UM cell lines, and could therefore potentially interfere with vasculogenic mimicry [147].Fasudil, a Rho kinase inhibitor, is able to reduce tumour growth in melanoma cell lines and melanoma mice models by inhibiting vasculogenic mimicry [148].Nicotinamide, the amide form of vitamin B3 (niacin), effectively targets vasculogenic mimicry by downregulating VE-cadherin in cutaneous melanomas cell lines. However, melanoma cells seemed to acquire an increased invasion capacity [149].A chemically modified tetracycline is able to inhibit MMP-2 and -9 in addition to laminin 5 in both cutaneous and uveal melanoma cell lines. Moreover, the expression of vasculogenic mimicry-associated genes is also inhibited [150].Cilenglitide, a potent inhibitor of αv integrins activation, reduces extracellular matrix invasion, vasculogenic mimicry, and secretion of MM9 by selectively targeting αvβ5 integrin in human cutaneous melanoma cell lines [151].Thalidomide, a drug with antiangiogenic and immunomodulatory properties, is able to decrease the number of vasculogenic mimicry tubules and reduce the protein expression of MMP-2, MMP-9, VEGF, proliferating cell nuclear antigen (PCNA), and nuclear factor-κβ (NF-κβ) in murine cutaneous melanoma cell lines [152].PARP inhibitors suppress the metastatic potential of some human and murine melanoma cells, due in part to the inhibition of vasculogenic mimicry mediated by the downregulation of VE-cadherin and the inhibition of the EMT pathway [153].Novel molecules, such as CVM-1118, seem to inhibit vasculogenic mimicry by targeting essential pathways involved in the process in human melanoma cells [154].


Targeting pericytes is another plausible therapeutic opportunity [155] and could help overcome tumour resistance to other drugs [156]. The proteoglycan NG2 stimulates the proliferation, motility, and migration of pericytes, and is crucial in the early stages of neovascularization [157]. Inhibition of pericytes via NG2 in UM xenografts decreases neovascularization and tumour volume, thereby rendering it a potential target [158,159].

Another possible approach would be a combination of antiangiogenic drugs with other therapies. For instance, combining bevacizumab with radiotherapy in UM mice models was shown to significantly decrease tumour growth compared to either bevacizumab or radiotherapy alone [160]. Combining antiangiogenic drugs with immunotherapy could have a potentially synergistic effect, as we will discuss in the following section.

## 4. The Potential Benefit of Combining Antiangiogenic and Immune Strategies

UM has traditionally been considered an immune privileged tumour, in a manner that closely parallels the microenvironment of the eye [161,162]. However, immune infiltration is found in the primary tumour [163,164], and its presence is associated to decreased survival [164], contrary to what is observed in other tumours [165]. Interestingly, metastatic UM seems to have a different lymphocytic composition compared to metastatic cutaneous melanoma, with predominant CD4+ lymphocytes instead of CD8+ lymphocytes (more commonly observed in cutaneous melanoma) [166]. Moreover, in contrast to most other tumours, the cytotoxic phenotype does not seem to confer better prognosis in UM [167], possibly reflecting the dominant immunosuppressive microenvironment [168,169]. UM also substantially differs from cutaneous melanoma in other aspects. For instance, the tumour mutation burden (TMB) (a surrogate for tumour antigenicity) of cutaneous melanoma is one of the highest among all tumours studied by the TCGA, whereas UM shows one of the lowest (Figure 3). However, when considering CD8A and PDL1 expression (a surrogate for tumour immunogenicity), almost 50% of cutaneous melanomas are considered CD8A^high^/PDL1^high^, whereas the same percentage of UM are considered CD8A^low^/PDL1^low^, reflecting a more immunosuppressed tumour microenvironment (Figure 4A,B). Interestingly the behaviour of UM and cutaneous melanoma in the highest and lowest CD8A/PDL1 quart differs substantially. Interestingly, CD8A^low^/PDL1^low^ cutaneous melanoma show a dismal prognosis, whereas CD8A^high^/PDL1^high^ UM convey a worse prognosis (Figure 4C).

A substantial number of UMs show no lymphocytic infiltration on histologic sections [164]. The immune infiltrate in UM seems to be related to genetic alterations, with the lack of BAP1 mutations showing a richer T-cell infiltration [170]. Another mechanistic explanation for this is the lack of adhesion of lymphocytes to the newly formed vessels [171]. Indeed, leukocytes require a number of molecules in order to roll, adhere, and finally transmigrate into the tumour microenvironment, including selectins, PECAM-1, intracellular adhesion molecular-1, -2 (ICAM-1, ICAM-2, respectively), and vascular cell adhesion molecular -1 (VCAM-1), amongst others [172]. Intratumoural cutaneous melanoma vessels have a decreased expression of P-selectin, VCAM-1, E-selectin, and ICAM-1, although the normal adjacent tissues have normal expression of these molecules [173,174,175,176]. The downregulation of these adhesion molecules could be mediated by the overexpression of VEGF [176]. Additionally, the tumour endothelium expresses molecules that can block T-cell infiltration in different tumour types (including melanoma), such as FasL [177], endothelin B receptor (ETBR), and endothelin-1 [178].

In addition to the expression of certain adhesion molecules on the endothelium, a chemoattractant is necessary to correctly recruit lymphocytes, especially chemokines that signal through the C-C chemokine receptor 5 (CCR5) and C-X-C Receptor 3 (CXCR3) axes [171]. Indeed, the co-expression of both molecules leads to a high CD8+ T-cell infiltration in cutaneous melanoma [179,180], and upregulation of both CCR5/CXCR3 is associated to greater response to different immunotherapies, including checkpoint inhibitors [181] and adoptive cell therapy [182]. Interestingly, the downregulation of these molecules or their corresponding ligands is correlated to disease progression [173,183].

Little is known about the interaction of vessels formed during vasculogenic mimicry and leukocytes. Nevertheless, studies have shown that the expression of some adhesion molecules on these vessels, such as PECAM-1 [128], possibly allowing circulating cells to interact with these tumour vessels. This raises the possibility of the tumour being able to regulate its own microenvironment by recruiting specific immune cells [171]. This area of research is currently under active investigation.

Highly angiogenic melanomas are more resistant to checkpoint inhibitors [38], probably due to the close relationship between aberrant cancer angiogenesis and immunosuppression [184]. Indeed, the tumour microenvironment, often characterized by hypoxia and high interstitial fluid pressure [185], could not only enhance an immunosuppressive microenvironment but also reduce the effectiveness of immunotherapy [37]. Moreover, VEGF (along with other angiogenic factors) plays a crucial role in modulating the immune system and fostering an immunosuppressive microenvironment: it directly suppresses dendritic cell maturation breast and colon carcinomas [39,186], inhibits T-cells by enhancing PD-1 and other inhibitory checkpoints in colon carcinomas [40,187], disrupts the normal differentiation of haematopoietic precursor cells [188], and recruits immunosuppressive cells such as T-cells [189] and myeloid derived suppressor cells [41,190].

Therefore, selectively targeting VEGF could not only inhibits angiogenesis but also change the tumour microenvironment, making it more “immunoresponsive” [185]. However, a more judicious use of antiangiogenic therapies would not only target angiogenesis but could give rise to tumour vessels with structural and functional phenotypes that are closer to non-malignant tissues, a process commonly referred to as vascular normalization [191]. This could subsequently result in an increased accumulation of cytotoxic T-cells [192], thereby improving the efficacy of checkpoint inhibitors.

The strategy of combining immune-checkpoint inhibitors with antiangiogenic drugs has been studied in other tumours [193]. For instance, three recent phase III studies combining anti-PD1/PDL1 with antiangiogenic therapies showed promising results in advanced renal cell carcinoma [194,195]. Pembrolizumab and axitinib increased overall survival compared to standard first-line sunitinib [195], and the combination of avelumab–axitinib increased progression-free survival [194]. Atezolizumab–bevacizumab also showed an increase in progression-free survival, although the survival data were still immature. In non-small cell lung cancer, the combination of chemotherapy with bevacizumab and atezolizumab increased progression-free survival and overall survival compared to chemotherapy and bevacizumab alone [196,197].

Based on these preclinical observations, Hodi et al. 2014 carried out a phase I clinical trial in patients with advanced cutaneous melanoma that received a combination of ipilimumab and bevacizumab. They observed a disease control rate of 64.7% and an OS of 25.1 months. More interestingly, however, on-treatment tumour biopsies revealed activated vessel endothelium with increased expression of E-selectin and increased infiltration of CD8+ T-cells [198]. Although angiogenesis seems to confer worse prognosis to UM when compared to cutaneous melanoma, to the best of our knowledge no clinical trial has been undertaken to study the benefits of combining antiangiogenic therapies with immunotherapy. The Grupo Español de Melanoma (GEM), a Spanish collaborative group, has recently designed a phase II, single-arm study in patients with metastatic UM who will be treated with the combination of durvalumab, an antiPD-L1 inhibitor, and cediranib, a multikinase inhibitor of VEGFR, PDGFRβ, KIT, and FLT-1 and -2. The primary endpoint is to evaluate the efficacy and response rate of the combination of cediranib and durvalumab in patients with metastatic UM with biopsiable disease at baseline, in first line or after failure to first-line systemic or liver-directed therapies. Five centres will be involved, and 18 patients are expected to be included, although the total number of patients may increase to 27 if the ORR > 20%.

## 5. Conclusions

In summary, angiogenesis plays an essential role in the development and progression of UM, and the exact implication of vasculogenic mimicry is still unclear, but could potentially play an important role. The efficacy of antiangiogenic drugs is still insufficient. Given the close relationship between angiogenesis and the immune microenvironment, combining antiangiogenic therapies and checkpoint inhibitors offers a potentially groundbreaking strategy. Nevertheless, we believe a deeper knowledge of vasculogenic mimicry is urgently needed. As we have seen, no effective strategy has been developed thus far and current antiangiogenic therapies not only do not target this pathway but could even overactivate it. This, in turn, could render antiangiogenic combination therapies ineffective. Moreover, targeting vasculogenic mimicry could not only contribute to the normalization of blood supply but also modulate the tumour microenvironment, therefore rendering immune therapies more effective.

## Figures and Tables

**Figure 1 cancers-11-00834-f001:**
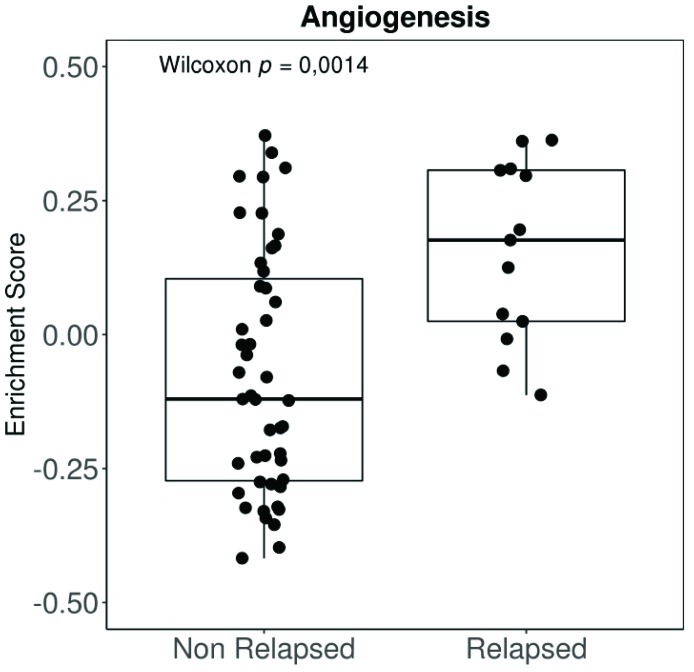
Angiogenesis enrichment score comparing relapsed vs. non-relapsed tumours. Scores were generated from expression data by gene set variation analysis (Gene Set Variation Analysis (GSVA) function). The gene set “Biocarta VEGF Pathway” from MSigDatabase (which includes genes related to hypoxia, blood vessels formation, and pro-angiogenic factors) was used (http://software.broadinstitute.org/gsea/msigdb/cards/BIOCARTA_VEGF_PATHWAY). Groups were compared by non-parametric Wilcoxon rank test. Data were extracted from The Cancer Genome Atlas (TCGA) database through cBioPortal (TCGA-UVM, 80 primary samples); expression data were downloaded as normalized Fragments Per Kilobase of exon per million fragments Mapped (FPKM) values and log2 transformed. VEGF: vascular endothelial growth factor.

**Figure 2 cancers-11-00834-f002:**
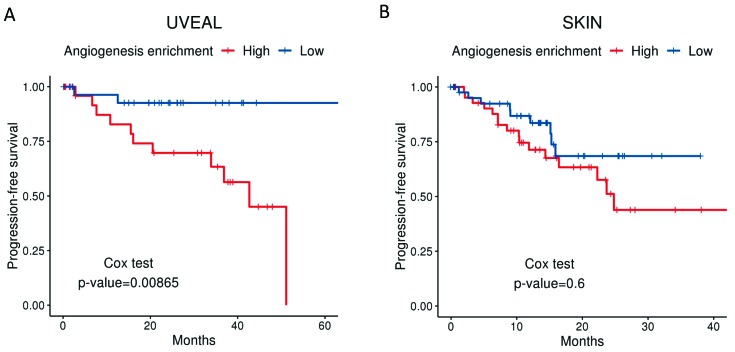
Disease-free survival for uveal melanoma (UM) (**A**) and cutaneous melanoma (**B**) from patients included in the TCGA, clustered into the top 50% (high angiogenesis enrichment score) vs. the bottom 50% (Low angiogenesis enrichment score). The cut-off used for generating high and low groups was the median enrichment score. The gene set used for scores was the same as for Figure 1. Data were extracted from the TCGA database through cBioPortal; 60 primary UM samples, 99 primary cutaneous melanoma samples.

**Figure 3 cancers-11-00834-f003:**
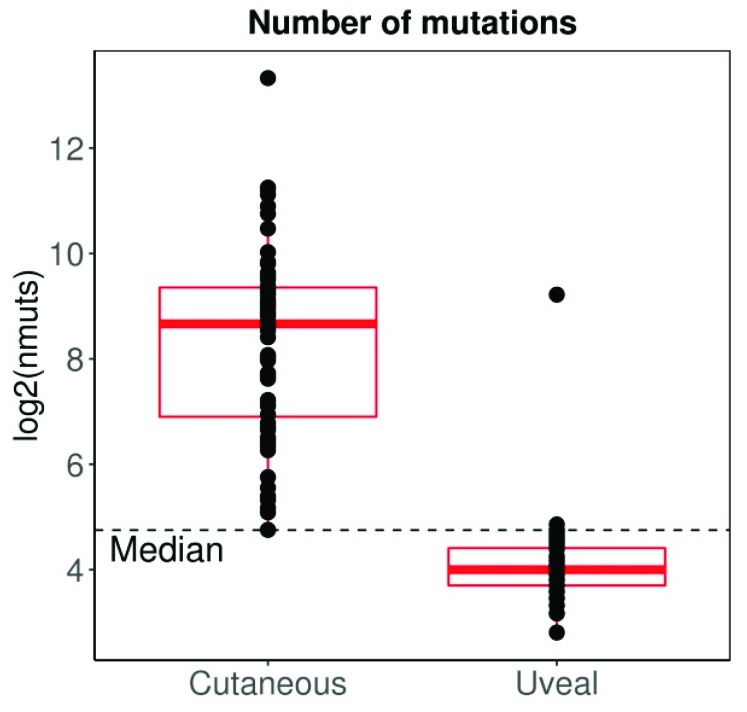
The mutation load of UM is in the lowest among all TCGAs, compared to cutaneous melanoma which has the highest. The median line has been calculated for all tumours included in the analysis. Data extracted from TCGA database through cBioPortal; 80 UM samples, 99 primary cutaneous melanoma samples. Expression data were normalized by TMM, log2 transformed, and scaled to allow proper comparison between datasets.

**Figure 4 cancers-11-00834-f004:**
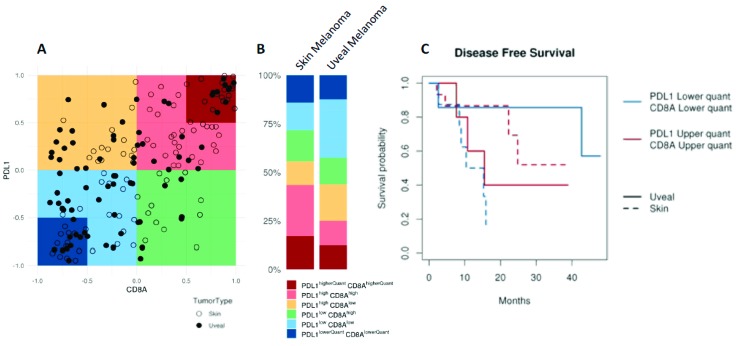
Comparison of immunogenicity between UM and cutaneous melanoma; Data extracted from the TCGA database. (**A**,**B**) Less than 25% of UM tumours express high CD8A and PDL1, whereas close to 50% of cutaneous melanoma tumour samples express high levels of both genes. (**C**) Kaplan–Meier PFS curves of UM and cutaneous melanoma with the highest (red) and lowest (blue) quartile of expression of both CD8A and PDL1. Highly immunogenic tumours seem to have better prognosis in cutaneous melanoma, although the opposite effect is seen in UM, even showing that all patients in the CD8A^high^/PDL1^high^-quartile experience disease relapse.

**Table 1 cancers-11-00834-t001:** Selected clinical trials with antiangiogenic drugs in metastatic uveal melanoma.

Author	Study	*N*	Regimen	Target	mOS (m)	mPFS (m)	ORR (%)	DCR (%)
Guenterberg 2011 [141]	Phase II	5	Bev-IFNα2b	VEGF (Bev, IFNα2b), bFGF (IFNα2b)	10.8	4.5	20	66
Tarhini 2011 [134]	Phase II	9	Aflibercept	VEGF-A, -B, PlGF	19	5.7	0	78
Zeldis 2009 [135]	Phase II	16	Lenalidomide	VEGF, others	NR	NR	0	44
Solti 2007 [136]	Pilot	6	Thalidomide-IFNα2b	VEGF, bFGF, TNFα	9	3.6	0	17
Bhatia 2012 [137]	Phase II	25	Carboplatin-paclitaxel + sorafenib	VEGFR-1–3, PDGFR-b, c-Kit, FLT-3, RET, Raf1, B-Raf	11	4	0	45
Mahipal 2012 [142]	Pilot	20	Sunitinib	VEGFR-1–3, PDGFR, c-Kit, FLT-3, RET	8.2	4.2	5	65
Hofmann 2009 [138]	Phase II	12	Imatinib	ABL, KIT, PDGFR	6.8	NR	0	8
Fruehauf 2011 [143]	Phase II	3	Axitinib	VEGFR-1, -2, -3	NR	NR	33	NR
Daud 2017 [139]	Phase II	23	Cabozantinib	c-Met, VEGFR-2, c-Kit, RET, FLT-3, TIE2, AxI	12.6	4.8	0	61
Piperno-Neumann 2013 [144]	Phase II	35	Bev-Temozolamide	VEGF	12	3	0	26
Scheulen 2017 [140]	Phase II	118	Sorafenib	VEGFR-1–3, PDGFR-b, c-Kit, FLT-3, RET, Raf1, B-Raf	14.8	5.5	1.7	67.8

Abbreviations: *N*: number of patients; mOS: median overall survival; mPFS: median progression-free survival; m: months; ORR: objective response rate; DCR: disease control rate; Bev: bevacizumab; IFN: interferon; PIGF: placental growth factor; VEGFR: vascular endothelial growth factor receptor; TNF: tumour necrosis factor.

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
