# Peer review of "Uveal Melanoma, Angiogenesis and Immunotherapy, Is There Any Hope?"

_cancers, 2019, doi:10.3390/cancers11060834_

Reviewer 1 Report

The review article by Piulats and co-workers comments on the progress of UM treatments based mainly on angiogenesis, and suggest the potential benefit of combining this approach with immunotherapy. Such combination is going to be tested soon by the Spanish Melanoma Group.

The article is very well presented and contains a lot of information well organized. However, for a review article, I recommend introducing some minor changes to make it more appealing to readers not just interested in UM. The article fits perfectly in the special issue on UM and should be accepted after some minor rearrangements

In this regard, a section (or a couple of sections) describing angiogenesis in general and then immunotherapy in general, should be included. The authors have described all the processes in relation to UM or Melanoma, however, a more general introduction to those approaches (with the latest strategies) including references for other tumors is desirable.

Furthermore, the section focused on angiogenesis is significantly larger than the section for immunotherapy, it would be better to balance a little both sections. The central hypothesis is the potential benefit of the combination of angiogenesis and immunotherapy, and despite the lack of clinical trials on UM using this approach, the authors should comment previous results in other diseases (not just melanoma) at the preclinical and clinical level to support this hypothesis.

Some reports that might be useful.

https://www.mdpi.com/1422-0067/18/11/2291

https://www.frontiersin.org/articles/10.3389/fimmu.2018.00527/full

https://link.springer.com/article/10.1007%2Fs10989-019-09860-0

Regarding the comment on the clinical trial sponsored by the GEM, it would be good if the authors could include more details of the study,… centers involved, end-point, some background on durvalumab and cediranib.

In page 9, line 293, “in dis matter” should be corrected by “in this matter”

Reviewer 2 Report

This is a well organised manuscript. Many interesting results are summarized from the literature. The new analyses performed by the authors increase the importance of the review. comparison of cutan and uveal melanomas provide new data at the field. 

Figure 4. organization of the text shoul be improve, The title is not really good for Figure 4.

Author Response

Se attached file

Reviewer 3 Report

This review provides very relevant information to understand angiogenesis and vascular mimicry in Uveal melanoma, as a basis for potential combination therapy. This is lovely work. However, one should also include less recent work, and any ref to the role of macrophages in vessel development and the genetic basis of inflammation in uveal melanoma is missing and should be added. The work of the groups of Grossniklaus, Kivela, Cree and Jager on angiogenesis is lacking and should be added.

And please clarify everywhere about which tumor you are writing: uveal melanoma, cutaneous melanoma, or other. Much of the work on vascular mimicry was NOT done on UM.

Abstract: I am not sure there is any data that show that vasculogenic mimicry is essential. It may only be a sign of high tumor aggressiveness, corresponding with loss of chromosome 3/class 2 tumors.

Minor;

When saying melanoma, always state whether it is cutaneous or uveal (or conjunctival)

Line 43; not Norway or Denmark. Choose one or say: and.

Line 49: name BAP1 as the last one, not the first. M3 and 8q were discovered much earlier.

Line 63; antibody instead of antibodies

Line 65: behind durabele complete… a word is missing

68: after CTLA-40 the word monoclonal antibody is missing

Lines 69 and 72: can you illustrate what you mean with 21 and 50%?

Line 109

There are many papers on the prognostic role of angiogenesis in uveal melanoma, and also about the role of macrophages that play a role in vessel development in uveal melanoma:

Foss AJ, Alexander RA, Jefferies LW, Hungerford JL, Harris AL & Lightman S (1996): Microvessel count predicts survival in uveal melanoma. Cancer Res 56: 2900-2903.

Makitie T, Summanen P, Tarkkanen A & Kivela T (1999): Microvascular density in predicting survival of patients with choroidal and ciliary body melanoma. Invest Ophthalmol Vis Sci 40: 2471-2480.

Makitie T, Summanen P, Tarkkanen A & Kivela T (2001): Tumor-infiltrating macrophages (CD68(+) cells) and prognosis in malignant uveal melanoma. Invest Ophthalmol Vis Sci 42: 1414-1421.

115: please explain the signature in the text. In fig 1, you name it the angiogenesis enrichment score, bit how the score works should be added to the paper. Which genes are used?

146; does NOTCH 1 play a role in uveal melanoma angiogenesis?

164-189 Please make it clear which of this work has been done in uveal melanoma and which work in other tumors.

191: which melanomas?

The statement in line 206 that in uveal melanoma, vasculogenic mimicry may play a predominant role in early stages is contradictec by the finding, that especially more malignant, larger, class 2 tumors have vasc mimicry (paper Maniotis and Harbour).

Line 208 that the highest OS rates have been reached with anti-angiogenics is an over statement: the number s of patients are so low, that no such statement can be made. Line 224 should state the group sizes.

Line 234 is also an overstatement to which hardly anyone would agree. When one looks at mPFD and ORR, the comment has to be changed.

The statement in line 243 that vas mimicry has been discussed and is essential, is NOT true. I see no evidence anywhere that it is essential. 249: which of these models used Uveal melanoma? From the text, it is not clear which melanoma was used, and which animal. Please clarify the texts.

Line 292 and elsewhere: immunosuppressive (2 ps!) Also line 299.

Line 292 add references to infiltrate

De Waard-Siebinga I.  HLA expression and tumor-infiltrating immune cells in uveal melanoma.

Graefe's Arch. Clin. Exp. Ophthalmol. 1996; 234: 34-42.

Line 293: this, not dis. (paragraph 4 should be checked for English)

Line 295: end of sentence is missing.

Fig 4C: have you not exchanged UM and CM?

UM usually die within 20 mths when highly inflamed.

317: the lack of lymphocytes in UM is genetically determined. Please use this information:

Ref. Gezgin G. Genetic evolution of uveal melanoma guides the development of an inflammatory microenvironment. Cancer Immunol Immunother 2017, 66 (7): 903-912. Doi: 10.1007/s00262-017-1991-1

327-333: it seems all of this refers to cutaneous melanoma. Please clarify.

340-349: it is here also not clear from which malignancies you get the statements.

Line 359: where do you get the statement from that UM is more dependent on angiogenesis than cutaneous melanoma?

Line 365 that recruitement is expected to start in the following months should be deleted, as this paper is going to be read in many years to come.

368: unless you have other statements to build on, change the sentence that vasculogenic mimicry is especially important in this tumour. Also downplay the statement that antiangiogeneic drugs seem to be more effective. I see no proof of that.

Do you know these papers?

Yang H., et al. Bevacizumab suppresses establishment of micrometastases of experimental ocular melanoma. Investigative Ophthalmology Vis Sci 2010; 51: 2835-2842.

Asnaghi L EMT-associated factors promote invasive properties of uveal melanoma cells.

Mol. Vis. 2015 Aug 25; 21: 919-929.

El Filali M., Anti-angiogenic therapy in uveal melanoma.

Dev. Ophthalmol. 2012, 49, 117-136.

Author Response

Se attached file

Round  2

Reviewer 3 Report

Excellent changes! I find this to be a very nice review which i will be very happy to share with my colleagues and students.

Should not the e mail addresses of all authors be shown?

Fig 4: several strange marks are shown in my pictures.

In the legend to fig 4 is an error; remove "are" before seem.

page 12/25: line 404: change despite to: Although, and say confer a worse prognosis to UM

line 418: remove clearly.

Author Response

Thank you for your constructive comments.

We have made all 4 changes you suggested and we have uploaded a new version of the manuscript.